# Differing Effects of Vinegar on *Pelagia noctiluca* (Cnidaria: Scyphozoa) and *Carybdea marsupialis* (Cnidaria: Cubozoa) Stings—Implications for First Aid Protocols

**DOI:** 10.3390/toxins13080509

**Published:** 2021-07-21

**Authors:** Ainara Ballesteros, Macarena Marambio, Verónica Fuentes, Mridvika Narda, Andreu Santín, Josep-Maria Gili

**Affiliations:** 1ICM-CSIC-Institute of Marine Sciences, Department of Marine Biology and Oceanography, Passeig Marítim de la Barceloneta 37-49, 08003 Barcelona, Spain; marambio@icm.csic.es (M.M.); veronica.cubomed@gmail.com (V.F.); santin@icm.csic.es (A.S.); gili@icm.csic.es (J.-M.G.); 2ISDIN, Innovation and Development, C. Provençals 33, 08019 Barcelona, Spain; mridvika.narda@gmail.com

**Keywords:** jellyfish, sting, first-aid, vinegar, seawater, nematocyst discharge, *Pelagia noctiluca*, *Carybdea marsupialis*

## Abstract

The jellyfish species that inhabit the Mediterranean coastal waters are not lethal, but their stings can cause severe pain and systemic effects that pose a health risk to humans. Despite the frequent occurrence of jellyfish stings, currently no consensus exists among the scientific community regarding the most appropriate first-aid protocol. Over the years, several different rinse solutions have been proposed. Vinegar, or acetic acid, is one of the most established of these solutions, with efficacy data published. We investigated the effect of vinegar and seawater on the nematocyst discharge process in two species representative of the Mediterranean region: *Pelagia noctiluca* (Scyphozoa) and *Carybdea marsupialis* (Cubozoa), by means of (1) direct observation of nematocyst discharge on light microscopy (tentacle solution assay) and (2) quantification of hemolytic area (tentacle skin blood agarose assay). In both species, nematocyst discharge was not stimulated by seawater, which was classified as a neutral solution. In *P. noctiluca*, vinegar produced nematocyst discharge *per se*, but inhibited nematocyst discharge from *C. marsupialis*. These results suggest that the use of vinegar cannot be universally recommended. Whereas in case of a cubozoan *C. marsupialis* sting, the inhibitory effect of vinegar makes it the ideal rinse solution, in case of a scyphozoan *P. noctiluca* sting, vinegar application may be counterproductive, worsening the pain and discomfort of the stung area.

## 1. Introduction

The Cnidaria phylum is a diverse group with more than 16,000 species distributed non-homogeneously around the world [1,2]. All its members, including jellyfish, share a distinctive feature: the presence of venom-filled stinging cells termed cnidocytes [3]. From the Golgi apparatus, a subcellular-enclosed capsule, known as a cnidocyst, is formed with an everted tubule along with a mixture of toxins [3,4]. Cnidocysts are classified into three categories: nematocyst, spirocyst and ptychocyst [3,5]. While the spirocyst and ptychocyst categories include only one cnidocyst type, the nematocyst category is more diverse [3]. The capsule size and shape, the length and pattern of the tubule (including absence or presence of spines), differ over thirty nematocyst types [3,6]. Following a chemical or mechanic stimulus, a rapid discharge process begins inoculating the venom through the tubule [4,7].

Jellyfish blooms are increasing in size and frequency in some areas of the world [8,9]. Human activities, such as overfishing, eutrophication, translocation, climate change and habitat modification stand out as the possible main causes of outbreaks [9]. Although jellyfish use nematocysts mainly for prey capture and defense against predators [3], 150 million jellyfish stings are reported worldwide annually in humans [10], being one of the most common reasons for seeking first-aid assistance from lifeguards during bathing seasons.

Spain is the country with the second-largest number of foreign visitors per year [11]. Most are attracted by the sun and sand model of the tourism industry, making beaches one of the region’s major assets [12]. However, beaches can become unsafe when there is high risk of exposure to jellyfish, with their known effect on human health [13,14,15,16]. This is especially true in the summer season, when an encounter between jellyfish and humans is more likely. In Spain, around 60% of the total first-aid encounters by lifeguards at beaches correspond to jellyfish stings [17]. At some beaches in the region of Catalonia, this number reaches 80% [18].

On the Spanish Mediterranean coast, species from the Scyphozoa class, such as *Pelagia noctiluca*, *Rhizostoma pulmo* and *Cotylorhiza tuberculata,* dominate [19]. Due to its wide distribution, abundance and the severity of its sting, *P. noctiluca* has become the most important jellyfish of the Mediterranean [19,20,21]. As such, this jellyfish is responsible for a high number of cases requiring assistance from rescue lifeguards [22,23] and has been classified as a highly venomous species due to the severity of its sting, which generally causes immediate pain, urticaria, edema, a burning sensation and formation of vesicles, papules and/or scabs [21,24,25]. *P. noctiluca’s* cnidome, the term for all types of cnidocytes within a cnidarian specimen including size, abundance and distribution during the entire life cycle, comprises four nematocyst types (a-isorhiza, A-isorhiza, O-isorhiza and eurytele) [26]. Despite the wide distribution of scyphozoans in the Mediterranean basin, the Cubozoa class is exclusively represented by *Carybdea marsupialis* [27,28]. Since the summer of 2008, the presence of this box jellyfish in some areas of the Spanish coast, like Denia (Alicante), has been a significant health problem, with thousands of sting-related cases requiring first-aid during the summer months [28]. Belonging to the most toxic class of cnidarians, *C. marsupialis* is not considered a lethal species, but its sting may produce painful skin lesions in the form of vesicles, red papules and/or burning. Systemic symptoms like heart and/or neurological complications have also been reported [25,29,30]. Nematocyst types such as isorhizas and eurytele are present in *C. marsupialis* adults [31].

The first-aid protocols for jellyfish stings define a set of actions that allow immediate assistance to those stung by jellyfish. Current protocols include the removal of jellyfish fragments attached to the skin, followed by washing the affected area with a rinse solution to help safely remove residual cnidocytes without nematocyst discharge. In the sting management scenario, avoiding a second envenomation episode is essential [16,25,32,33].

As of today, there is no consensus on the most appropriate rinse solution to wash the affected area, with wide debate among the scientific community. Recommendations include seawater, vinegar, urine and alcohol, with vinegar (or acetic acid) being the most studied and the one that generates the greatest debate [16,25,32,33]. While some research groups support the universal use of vinegar regardless of the species type [34], others suggest re-evaluation of the currently recommended rinse solutions [16] and more research to discover new compounds [32]. However, recommendations must take into account the differences in nematocyst types and the different possible responses to the same stimuli. Considering this, prior to their recommendation as a post-sting rinse solution, such solutions ought to be subjected to species-specific tests to determine their efficacy, avoiding the extrapolation of results that may cause bad practices in first-aid protocols.

The objective of the present study was to determine (1) the effect of vinegar and seawater rinse solutions on the nematocyst discharge process using a tentacle solution assay (TSA) methodology and (2) the inhibitory effect of vinegar on nematocyst discharge (TSA) and venom load using the tentacle skin blood agarose assay (TSBAA) method in *P. noctiluca*, the dominant venomous species, and *C. marsupialis,* the only jellyfish of the Cubozoa class present on the Mediterranean coast.

## 2. Results

### 2.1. Pelagia noctiluca

#### 2.1.1. Test 1. Rinse Solution Screening

Nematocyst discharge—tentacle solution assay (TSA). Nematocysts were not activated after incubation in seawater (Figure 1A and Table 1). As such, seawater was classified as a neutral solution (Table 1). Conversely, nematocyst discharge was observed after incubation of the tentacle with vinegar (Figure 1B–D). All nematocysts, including a-isorhiza, A-isorhiza, O-isorhiza and eurytele (Figure 1B,C) were discharged. Therefore, vinegar was classified as an activator solution (Table 1).

#### 2.1.2. Test 2. Evaluation of the Inhibitory Effect

Natural stimulation of nematocyst discharge—tentacle skin blood agarose assay (TSBAA). As vinegar was an activator solution for *P. noctiluca* (Figure 1B–D) (Section 2.1.1), inhibition testing using chemical nematocyst stimulation (TSA) was not performed. However, its effectiveness was tested using TSBAA to determine if any changes in venom load were produced despite the discharge. After the natural sting process, the hemolytic area was 23.80 ± 06.90% with seawater and 21.64 ± 11.98% with vinegar (Figure 2 and Table 2). The statistical analysis revealed no significant difference between both rinse solutions and, therefore, vinegar did not reduce the hemolytic capacity of *P. noctiluca* venom (Table 2).

### 2.2. Carybdea marsupialis

#### 2.2.1. Test 1. Rinse Solution Screening

Nematocyst discharge—tentacle solution assay (TSA). Nematocyst discharge was not visible after seawater or vinegar incubation (Figure 3A,C). In this first phase, both rinse solutions were classified as neutral solutions (Table 1).

#### 2.2.2. Test 2. Evaluation of the Inhibitory Effect

Chemical stimulation of nematocyst discharge—tentacle solution assay (TSA). For *C. marsupialis*, in contrast to *P. noctiluca*, the inhibitory effect could be tested using an activator solution *per se* (ethanol) (Figure 3B and Table 3) since, after the vinegar incubation, the nematocysts remained intact, demonstrating it was a neutral solution (Section 2.2.1) (Table 1). In the case of seawater, nematocysts were activated after the application of ethanol. Seawater was classified as a neutral solution (Table 3). In contrast, vinegar inhibited the nematocyst discharge process completely despite chemical stimulation (Figure 3D). No empty capsules or discharged tubules were observed. Vinegar was considered an inhibitor rinse solution (Table 3).

Natural stimulation of nematocyst discharge—tentacle skin blood agarose assay (TSBAA). After incubation in seawater and the sting process, the hemolytic area observed obtained a value of 32.49 ± 19.33%. Seawater was classified as a neutral solution (Table 2). However, the hemolytic area data (0.73 ± 2.31%) reflected the inhibitory effect of vinegar (Figure 2 and Table 2) preventing nematocyst discharge and reducing hemolysis almost entirely (Table 2). The statistical analysis revealed significant differences between both rinse solutions (Table 2).

## 3. Discussion

Due to the enormous health and socio-economic impact of jellyfish stings in coastal areas, identifying solutions to effectively inhibit nematocyst discharge is of great importance as these can then be employed to rinse the affected area post-sting. Publications suggest the use of vinegar as an ideal rinse solution to safely remove residual jellyfish tissue, without risk of further activation of remaining cnidocytes, indiscriminately across different classes of the Cnidaria phylum (Cubozoa, Hydrozoa and Scyphozoa) [34]. However, some previous studies have shown that the nematocyst discharge response may differ between different jellyfish species or even classes [35,36].

In the present study, we compared the effect of vinegar on nematocyst discharge in two representative jellyfish of different classes; *P. noctiluca* (Scyphozoa) and *C. marsupialis* (Cubozoa), both found in the Mediterranean Sea along the Spanish coast [19,27,28]. In the current study, vinegar was identified as an inhibitor of nematocyst discharge for *C. marsupialis* as it successfully inhibited discharge even when discharge had been induced by an activator solution (Figure 3D and Table 3). In contrast, it was identified as an activator *per se* for *P. noctiluca* nematocysts (Figure 1B–D and Table 1).

Previous research did not show nematocyst discharge after the addition of vinegar or acetic acid for cubozoan jellyfish, generating an inhibitory effect or a significant reduction in the venom load [35,36,37,38,39,40], as we also found for *C. marsupialis*. However, nematocyst discharge and exacerbation of pain are well-known in the presence of vinegar or acetic acid for the Scyphozoa class [35,36,41]. Contrary to our results, Morabito et al. (2014) [42] concluded that 5% acetic acid was effective as an inhibitor solution for *P. noctiluca,* recommending it as a local treatment for stinging. However, our work shows that the vinegar caused immediate nematocyst discharge and created a hemolytic area due to the venom penetrating into the SRBC agarose (Figure 1B–D and Figure 2 and Table 2). The differences in the results between the two studies can be attributed to the methodology used. In the study by Morabito et al. (2014) [42], nematocyst discharge was induced by a combined physico-chemical stimulation by chemo-sensitizers in the presence of 5% acetic acid, followed by mechanical stimulation with a non-vibrating test probe. In contrast, we used TSA methodology, a simple technique that is widely used all over the world [34,35,36,37,38,39,40]. The observation of the direct effect of the rinse solution on the discharge of nematocysts avoids possible methodological errors. Moreover, in accordance with our findings (Table 1), nematocyst discharge for *P. noctiluca* had been reported previously with extreme acidic pH values [43]. Since our TSA results were later corroborated by the TSBAA methodology (Table 1 and Table 2), we conclusively state that the use of vinegar or acetic acid after a *P. noctiluca* sting can lead to further nematocyst discharge and is clearly detrimental.

A recent study with the scyphozoan *Cyanea capillata* [34] concluded that vinegar caused an almost complete decrease in venom activity by TSBAA methodology despite its direct nematocyst discharge stimulation by the TSA method. The authors suggested that these opposite results could be attributed to the non-discharge of A-isorhiza and O-isorhiza nematocysts, which are the types that contribute most to hemolytic activity. Activation of A-isorhiza and O-isorhiza in the presence of acetic acid was formerly reported for scyphozoan species [36,44], including *C. capillata* [44]. Our study supports the discharge of A-isorhiza and O-isorhiza types for scyphozoan jellyfish (Figure 1B,C).

The presence of a physical barrier (pig intestine) in the TSBAA methodology was a modification proposed by Yanagihara et al. (2016) [39] to avoid solution-induced hemolysis or effect-masking color change. The pig intestine was pretreated with anhydrous lanolin to prevent leaks, which can occur particularly in the presence of vinegar. Pretreating the pig intestine skin in this way can reduce leaks, but the lanolin can in turn act as a physical barrier, preventing the penetration of nematocyst tubules into the SRBC agarose. From our experience, we recommend working without lanolin in TSBAA experiments whenever possible; if necessary, a thin layer should be used to ensure that the reduction in the hemolytic area is only due to the effect of the solution on the venom load.

Seawater did not stimulate the discharge process in scyphozoan and cubozoan species [34,35,36,39,40]. Its neutral effects have been confirmed for *P. noctiluca* and *C. marsupialis* (Figure 1A and Figure 3A). Because of its neutral condition, some authors have suggested that seawater is not a good recommendation to wash the sting area, as nematocysts roll off the skin and discharge may occur due to mechanical effects [34,39,40]. Nevertheless, seawater reduced pain and redness for stings from the scyphozoan and cubozoan species [35,36]. Due to its unclear efficacy, it is difficult to conclude the implications of seawater in first-aid protocols.

All the evidence described above reflects the variability of the response of nematocyst discharge in the presence of vinegar. The activation of *P. noctiluca* nematocysts prevents the universal use of vinegar and opens new lines of research to explore new rinse solutions that may be used universally in all species. Such a solution is possible if its efficacy can be demonstrated in different classes or species but, if not, species-specific first-aid protocols must be established.

## 4. Conclusions

Jellyfish stings account for a large percentage of first-aid cases on beaches. To minimize the adverse effects from jellyfish stings, it is important to establish first-aid protocols based on scientific evidence and improve them to create species-specific protocols as required.

Our research shows that the response of the nematocyst discharge process differs between classes of the phylum Cnidaria. Different species of jellyfish respond differently to vinegar rinse solution. This species specificity must be kept in mind when developing first-aid protocols for jellyfish stings. Vinegar has demonstrated efficacy for inhibiting nematocyst discharge in cubozoan species. However, our research has shown that vinegar, or commercial products based on this compound, cannot be recommended for stings related to *P. noctiluca*, the predominant venomous species found on the Spanish Mediterranean coast. Its use may be counterproductive, causing direct discharge of the nematocysts, which may worsen pain in the affected person.

We recommend the use of vinegar as an inhibitor rinse solution for *C. marsupialis*. For the treatment of *P. noctiluca* stings, the use of seawater is suggested due to its neutral effect, until a solution with an inhibitory effect can be identified.

## 5. Materials and Methods

### 5.1. Collection and Preparation of Jellyfish

*P. noctiluca* individuals were collected in open waters adjacent to Barcelona (Catalonia, Spain) during November 2018 and *C. marsupialis* individuals were collected along the Denia coast (Alicante, Spain) during September 2018. Both samplings were performed at night. In the case of cubozoans, a light was used to attract them. Individuals were collected using hand nets and plastic jars to avoid tissue damage. The specimens were kept in plastic bags with seawater, avoiding air bubbles inside, to ensure optimal transport conditions. The experiments were conducted in the Institute of Marine Sciences (ICM-CSIC) in Barcelona (*P. noctiluca*) and Montgó-Dénia Scientific Station (ESCIMO-Dénia) (*C. marsupialis*). The day after sampling, the tentacles of both species were cut using dissection scissors and stored in containers with micro-filtered seawater at 4 °C until their use (less than 48 h).

### 5.2. Rinse Solutions and pH Measurements

The rinse solutions were seawater (control) and vinegar (Vivó, 6% acetic acid). The pH value of each solution was measured with a benchtop pH meter (Orion Star A211).

### 5.3. Test 1. Nematocyst Discharge—Rinse Solution Screening

Method 1. Tentacle solution assay (TSA) [39].

Tentacle pieces of approximately 3 cm long were transferred to slides (76 × 26 cm). The samples were observed under a light microscope to ensure their integrity. Subsequently, tentacles were incubated in microwells with 3 mL of seawater or vinegar for 5 min. The preparation was covered carefully with the coverslip and placed under the light microscope to observe the nematocyst response.

The nematocyst response was classified qualitatively into four categories in accordance with Pyo et al. (2016) [36]:0: no discharge was observed;+: low discharge of nematocysts;++: medium discharge of nematocysts;+++: high discharge of nematocysts.

The solutions were classified into one of two categories, discarding activator solutions *per se*:Activator solution: nematocysts were activated after incubation in the solution;Neutral solution: nematocysts were not activated after incubation in the solution.

### 5.4. Test 2—Evaluation of the Inhibitory Effect

#### 5.4.1. Chemical Stimulation of Nematocyst Discharge

Method 1. Tentacle solution assay (TSA) [39].

After screening of substances, activator solutions, which produced nematocyst discharge, were discarded. Only neutral solutions (Section 5.3) were evaluated.

To determine the inhibitory effect in *C. marsupialis*, nematocyst discharge was chemically stimulated with ethanol (PANREAC, ethanol 96%).

After the seawater and vinegar incubations (Section 5.3), 15 µL of ethanol were applied to the tentacle.

The solutions were classified into one of two categories:Neutral solution: nematocysts were not activated after incubation in the first solution but were activated by the subsequent chemical stimulation with ethanol solution;Inhibitor solution*:* nematocysts were not activated after incubation in the first solution or by the subsequent chemical stimulation with ethanol solution.

#### 5.4.2. Natural Stimulation of Nematocyst Discharge

Method 2. Tentacle skin blood agarose assay (TSBAA) [39].

The venom load following contact with the tentacle was evaluated by quantification of the hemolytic area in a modified protocol adapted from the TSBAA method published by Yanagihara et al. (2016) [39]. Briefly, an agarose gel preparation incorporating sheep red blood cells (SRBCs) (Thermo Fisher Scientific) was used, covered by a thin tissue of pig intestine to simulate the effect of the human skin barrier.

A solution of 4% low-melting-point agarose (INVITROGEN ™ LMP Agarose) in phosphate buffered saline (PBS) (SIGMA) was prepared and cooled to room temperature for a few minutes. The SRBCs were centrifuged and resuspended in PBS until a final concentration of 2% was obtained. Equal volumes of both preparations were mixed and poured onto a glass surface with a mold (18 × 7.5 cm) and allowed to cool to room temperature. Once the agarose with SRBCs was set, rectangles were cut and placed on slides and kept at 4 °C in a humidification chamber (maximum 72 h until use).

Pig small intestine tissue was washed in a 50 mM saline solution and then cut in rectangular sections. The sections were sterilized in a disinfecting solution consisting of 87% 110 mM saline, 10% ethanol (PANREAC, ethanol 96%) and 3% hydrogen peroxide for 3 h, after which they were left in physiological serum overnight. The next day, the sections were stretched on a glass surface and allowed to dry for a few minutes in the open air. The pretreated sections of the pig small intestine were placed on the SRBC agarose rectangles; a thin layer of anhydrous lanolin for pharmaceutical use was applied.

Based on the results from the rinse solution screening using TSA assay (Section 2.1.1 and Section 2.2.1), one of the following approaches was used for the venom-loading experiment:Neutral solution: tentacle was incubated in microwells for 5 min to carry out the stinging process after incubation.Activator solution*:* solution was applied in spray format on the top of the pig intestine to avoid nematocyst discharge into the microwells during the incubation period.

In order to simulate the sting process in a natural way, the surface of the molds was exposed to approximately 2 cm of *C. marsupialis* tentacle, in a linear arrangement, and 3 cm of *P. noctiluca* tentacle, in a circular arrangement, for 2 min. The circular presentation for *P. noctiluca* was chosen to maximize the venom at one point due to their lower toxicity compared to *C. marsupialis*.

After the sting event, the intestine sections were removed and the SRBC agarose was stored in the humidification chamber at room temperature. After a period of 22 h, images of the hemolytic areas were obtained with a camera. Hemolytic areas were calculated using the Fiji version of ImageJ software [45]. The data was tested for normality and homogeneity using the *stats* package available for the R software platform [46]. As data did not display homogenous variances, a Mann–Whitney test was applied by means of the *wilcox.test* function to test if significant differences existed between hemolytic areas. Finally, a graphical representation was performed using the *ggplot2* package from the R software platform [47].

## Figures and Tables

**Figure 1 toxins-13-00509-f001:**
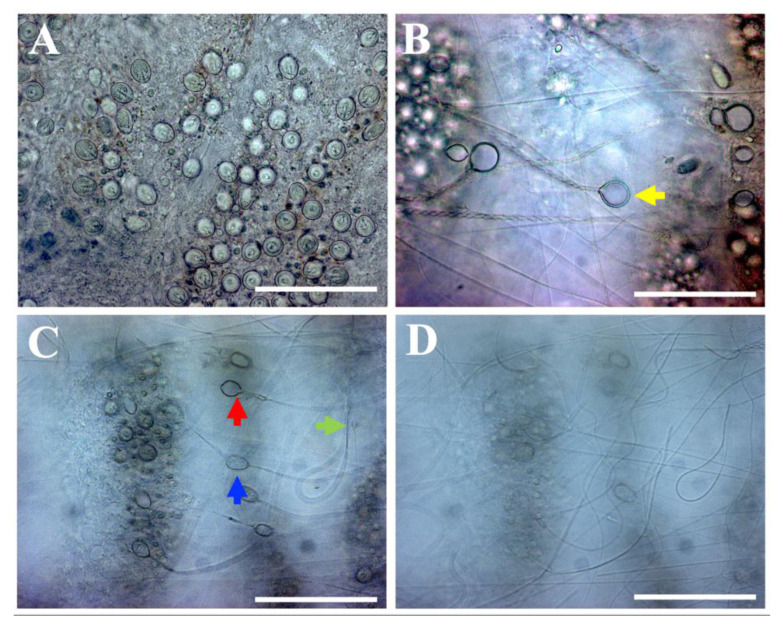
Nematocyst response of *P. noctiluca* after incubation in: (**A**) seawater and (**B**–**D**) vinegar. Note in the presence of vinegar: (**B**) discharged O-isorhiza (yellow arrow); (**C**) discharged a-isorhiza (green arrow), A-isorhiza (blue arrow) and eurytele (red arrow); and (**D**) all tubules discharged. Scale bars: 100 µm.

**Figure 2 toxins-13-00509-f002:**
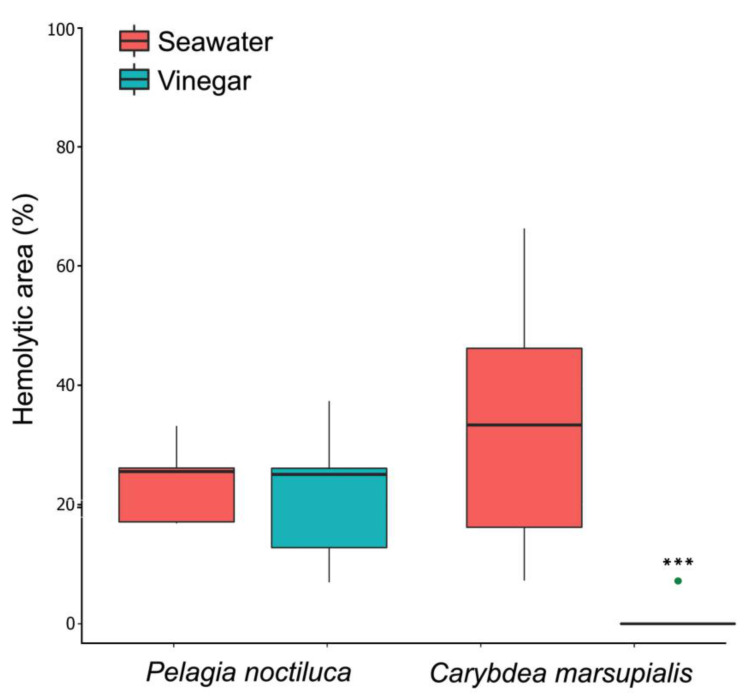
Hemolytic area (%) produced by the venom using TSBAA. Note the inhibitory effect of vinegar in the case of *C. marsupialis* (*** *p* = < 0.001) (Mann–Whitney test). Box plots with bottom and top of the box as first and third quartiles and horizontal line inside the box as second quartile (median). Outlying data is plotted as green dot.

**Figure 3 toxins-13-00509-f003:**
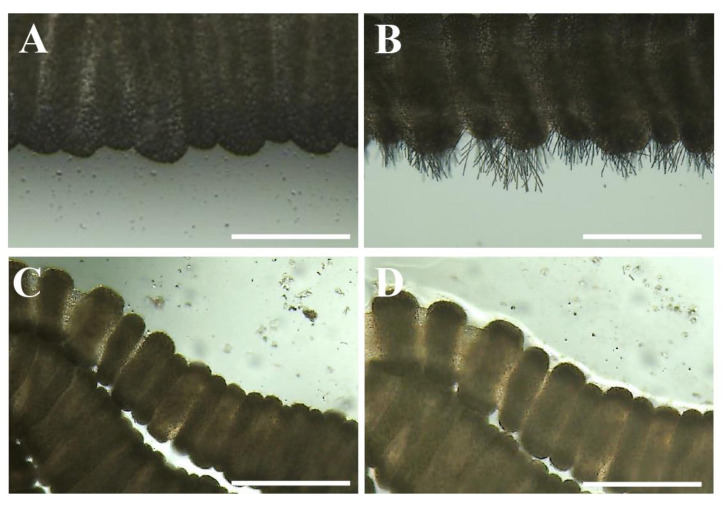
Nematocyst response of *C. marsupialis* after incubation in: (**A**) seawater, (**B**) ethanol and (**C**) vinegar. Note the different response between a neutral solution (**A**) and an activator solution *per se* (**B**). (**D**) Inhibitory effect after a first incubation in vinegar and subsequent application of ethanol. Note no nematocyst discharge despite chemical stimulation by ethanol. Scale bars: (A, B) 0.5 mm, (C, D) 1 mm.

**Table 1 toxins-13-00509-t001:** Rinse solution screening using TSA after incubation in seawater or vinegar, with pH values.

		*Pelagia noctiluca*			*Carybdea marsupialis*		
Rinse Solution	pH	*n*	Discharge ^1^	Effect ^2^	*n*	Discharge ^1^	Effect ^2^
Seawater (control)	7.80	6	0	Neutral	3	0	Neutral
Vinegar	2.60	6	++	Activator	3	0	Neutral

Method 1. Tentacle solution assay (TSA). ^1^ Nematocyst discharge categories: 0 = no discharge was observed; ++ = medium discharge of nematocysts; ^2^ Rinse solution categories: activator solution = nematocysts activated after incubation in the solution; neutral solution = nematocysts not activated after incubation in the solution; *n* indicates the number of replicates.

**Table 2 toxins-13-00509-t002:** Inhibitory effect of vinegar after natural stimulation using TSBAA.

	*Pelagia noctiluca*			*Carybdea marsupialis*		
Rinse Solution	*n*	Hemolytic Area (%)	Effect	*n*	Hemolytic Area (%)	Effect
Seawater (control)	5	23.80 ± 6.90	Not inhibitor	10	32.49 ± 19.33	Not inhibitor
Vinegar	5	21.64 ± 11.98 ^ns^	Not inhibitor	10	0.73 ± 2.31 ***	Inhibitor

Method 2. Tentacle skin blood agarose assay (TSBAA). The inhibitory effect of vinegar was tested using a Mann–Whitney test for non-parametric data, results being expressed as follows: ns: *p* > 0.05 (not significant), *** *p* = < 0.001. *n* indicates the number of replicates.

**Table 3 toxins-13-00509-t003:** Inhibitory efficacy of vinegar after chemical stimulation using TSA.

		*Carybdea marsupialis*	
Rinse Solution	*n*	Discharge ^1^	Effect ^2^
Ethanol	3	+++	Activator
Seawater (control) + ethanol	3	+++	Neutral
Vinegar + ethanol	3	0	Inhibitor

Method 2. Tentacle solution assay (TSA). ^1^ Nematocyst discharge categories: 0 = no discharge was observed; +++ = high discharge of nematocysts; ^2^ Rinse solution categories: activator solution = nematocysts activated after incubation in the solution; inhibitor solution = nematocysts not activated after incubation in a first solution nor by the consecutive stimulation by a known activator (ethanol). *n* indicates the number of replicates.

## Data Availability

Data sharing not applicable. No new data were created or analyzed in this study.

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
