# Peer review of "Differing Effects of Vinegar on Pelagia noctiluca (Cnidaria: Scyphozoa) and Carybdea marsupialis (Cnidaria: Cubozoa) Stings—Implications for First Aid Protocols"

_toxins, 2021, doi:10.3390/toxins13080509_

Round 1

Reviewer 1 Report

The study is concerned with the question which rinse solution should be used to treat jellyfish stings without triggering the discharge of further capsules. To address this in a species-specific way the authors used two common jellyfish from the Mediterranean (P. noctiluca and C. marsupialis) and tested the effect of vinegar as it is the most widely used rinse solution.

The effect of vinegar is assessed in two different experiments:

1) Direct observation of the capsule discharge after rinsing with seawater, vinegar and following chemical activation by ethanol

2) Observation of hemolytic areas 22h after a jellyfish sting in an agar-based hemolysis assay

The results essentially show that vinegar activates the discharge of capsules P. in noctiluca leading to a discharge of all capsule types but to no significant hemolysis (Exp 2).

In C. marsupialis on the other hand vinegar efficiently inactivates the capsules leading to no visible discharge of capsules and a significantly reduced hemolytic area (Exp2).

The authors conclude that vinegar can be used to rinse C. marsupialis stings but might lead to adverse effects in other jellyfish and should therefore not be used indiscriminately for every jellyfish sting. They also propose that new rinse solutions should be tested using different species of jellyfish in the future.

Overall the study is very much in line with a previous, more extensive report by Pyo et al., 2016. The novelty of the work is therefore, apart from the use of different jellyfish species, quite limited.

Specific points:

  1. Please rectify throughout the text: stinging cells are called cnidocytes (or nematocytes), the stinging organelles are called cnidocysts (or nematocysts).

  1. References on general nematocyst biology are outdated and incomplete.

  1. The applied vinegar is not accurately defined. Does it have any ingredients (like stabilizers) that might affect nematocyst discharge? Does it match the rinse solution commonly used by life guards?

  1. It would be interesting to test nematocyst discharge at different pH conditions in a systematic manner.

  1. Nematocyst discharge is calcium-dependent process. Have the authors considered to apply EDTA or similar chelators in rinse solutions? Also, several studies have shown that the nematocyst capsule wall structure is sensitive to reducing agents like DTT. This might be an even more effective and general way to inhibit further discharge.

  1. 1 B-D shows mostly discharged cnidocysts disconnected from the tentacle tissue, while Fig. 1A shows non-discharged cnidocysts still embedded in tissue. This is somewhat misleading as tissue preparations often produce detached capsules that discharge more easily than those still embedded in the tentacle.

  1. I recommend labeling the images directly with the species and rinse solution used.

  1. Figure 2: What is meant by “..pass of ethanol through the tentacle”? It looks like as if the assignment of scale bars in the legend got mixed up. I assume that it should be 0.5 mm in A + B and 1 mm in C + D as these images seem to have a lower magnification.

  1. Figure 3: please explain why Fig 3 D is blue and the other vinegar treatment in Fig 3B is red, although lanolin and pig intestine was used to prevent discolorations.

  1. The text needs thorough proofreading by a native speaker.

Reviewer 2 Report

This manuscript presents a short study aiming at testing the effect of vinegar on the stings of two Mediterranean jellyfish. The authors show that vinegar has different and even opposite effects when applied to the tentacles of Pelagia noctiluca and Carybdea marsupialis.  The results provided are modest but novel. I have only minor comments:

- Line 12 of the abstract: correct “marsupilais”

- Line 3 of the main text: “The distinctive cells responsible for the cnidarian stings are called cnidocysts” Here it should be cnidocytes and not cnidocysts since we talk about the cells and not the capsule.

- Cnidocyst and nematocyst are used interchangeably in the manuscript without stating that they are synonymous. Please either state that these two terms are synonymous or use only one of the two throughout the manuscript.

- The meaning of the abbreviations TSA and TSBAA is not explained in the result part, only in the materials and methods section. This makes several of the experiments difficult to understand for readers non familiar with these types of assays (especially those presented in 2.2). I would suggest briefly explaining what those two methods are, notably by expending the introductory paragraph of the section 2.2.

- Figure 3 should be improved. It is unclear what is shown in those images and the figure legend does not provide much information. Scale bars are also missing. I am also wondering why the background color is different between A-C and D. Please provide additional explanations in the figure legend, as well as labelling on the images and/or better images..

- Section 2.2, second line. “the the”

Round 2

Reviewer 1 Report

I'm ok with the revised manuscript. Minor point: please specify what the orange arrow in Fig. 3D is meant to highlight. Was the ethanol applied only locally at the place where the arrow is pointing at? If not, it is irritating to have an arrow in this figure.